# Microneedle Mediated Gas Delivery for Rapid Separation, Enhanced Drug Penetration, and Combined Therapy

**DOI:** 10.3390/pharmaceutics17121576

**Published:** 2025-12-07

**Authors:** Ziyang Zheng, Ting Zhou, Hongluo Li, Jade Jillian Xian Lan Zeng, Yanping Fu, Chao Lu, Tingting Peng, Chuanbin Wu, Guilan Quan

**Affiliations:** 1State Key Laboratory of Bioactive Molecules and Druggability Assessment, Guangdong Basic Research Center of Excellence for Natural Bioactive Molecules and Discovery of Innovative Drugs, College of Pharmacy, Jinan University, Guangzhou 511436, China; 18328305097zzy@stu.jnu.edu.cn (Z.Z.); zhouting@stu2024.jnu.edu.cn (T.Z.); fuyanping@stu2021.jnu.edu.cn (Y.F.); chaolu@jnu.edu.cn (C.L.); pengtt@jnu.edu.cn (T.P.); chuanbinwu@jnu.edu.cn (C.W.); 2School of Automobile and Transportation Engineering, Guangdong Polytechnic Normal University, Guangzhou 510665, China; 3International School, Jinan University, Guangzhou 511436, China; jadezeng@stu2023.jnu.edu.cn

**Keywords:** microneedles, rapid separation, drug penetration, gas therapy, combined therapy

## Abstract

In recent years, microneedles (MNs) have emerged as a novel transdermal drug delivery technology, offering advantages such as avoidance of the first-pass effect, pain-free and minimally invasive administration, and convenient application. However, conventional MNs still face challenges, including slow detachment of MN tips from the base substrate and limited transdermal efficiency. This review systematically summarizes recent advances in MNs-mediated gas delivery for rapid separation, enhanced drug penetration, and combined therapy. The discussion encompasses the benefits and limitations of MNs and recent developments in MN-facilitated gas delivery to accelerate separation rate and improve delivery efficiency. By analyzing the therapeutic roles of various gases (e.g., H_2_, O_2_, NO, H_2_S, CO, CO_2_) and their synergistic potential when combined with MNs, this review also provides insights and references for the further application of gas-assisted MN systems for combined therapy in various disease treatments.

## 1. Introduction

Transdermal drug delivery systems (TDDS) are a method of administering medications through the skin. The drug is absorbed through the skin’s blood vessels into the body’s circulation and subsequently enters into the systemic circulation. In addition, these systems encompass formulations that facilitate localized therapeutic responses. In comparison with conventional drug delivery methodologies, TDDS offers several advantages, including the circumvention of hepatic first-pass metabolism, reduced invasiveness, ease of application, and the elimination of the necessity for specialized medical personnel [1]. However, the efficacy of TDDS is significantly hindered by the innate skin barrier, particularly the stratum corneum, which limits drug penetration. To address this challenge, current studies are exploring a variety of strategies. These include the formulation of drugs into micro- and nano-sized carriers [2], as well as chemical enhancement methods including chemotactic agents, liposomes, and advanced nanoformulations. Physical enhancement techniques such as iontophoresis, ultrasound, electroporation, and thermal ablation are also being explored [1]. However, these methods impose stringent requirements on the physicochemical properties of the drug, exhibit skin irritation, and suffer from the drawback of being equipment-dependent [1].

This review focuses specifically on microneedles (MNs) due to their ease of fabrication, reliability, and ability to effectively deliver therapeutic doses [3]. MNs are composed of hundreds of microscopic needles arranged on small patches, and they are structurally similar to conventional transdermal patches [4]. MNs can penetrate the stratum corneum and create microchannels within the skin. This process has been demonstrated to remarkably enhance drug penetration efficacy while maintaining minimal invasiveness [5]. Consequently, MNs have demonstrated great potential for treating a variety of medical conditions, including skin disorders [6,7], neurological diseases [8,9], bone diseases [10,11], cardiovascular diseases [12,13,14], and endocrine diseases [15,16,17,18]. Furthermore, the integration of wearable sensors with MNs has led to advancement in real-time detection and drug delivery, enabling precise and on-demand medication administration [19,20].

However, MN-based delivery methods encounter significant challenges. Firstly, the hydrophilic polymer-based MNs exhibit a slow dissolution rate in the skin compared to in vitro conditions, resulting in insufficient needle tip-substrate separation [21]. This results in prolonged application times and reduced patient compliance. Secondly, the transdermal efficiency of MNs remains limited, primarily due to passive diffusion of drugs through multiple skin layers, which restricts deep penetration in dense lesions and systemic absorption [22]. Thus, developing MNs that accelerate separation and enhance skin penetration is crucial.

Recently, gas therapies employing bioactive gaseous molecules (e.g., nitric oxide (NO), oxygen (O_2_), hydrogen (H_2_), hydrogen sulfide (H_2_S), sulfur dioxide (SO_2_), and carbon monoxide (CO)) have emerged as promising therapeutic adjuncts [23]. These gases possess therapeutic properties and can enhance drug diffusion and release via gas dynamics, thereby improving drug bioavailability [24]. The integration of MN delivery technology with gas therapies has the potential to leverage MN’s drug delivery advantages, thereby addressing current limitations such as slow MN separation rates and low transdermal efficiency. This synergistic approach integrates the enhanced delivery efficiency of MNs with the therapeutic benefits and dynamic augmentation of gas therapies.

Overall, in comparison with conventional transdermal technologies, MNs offer the advantage of creating micro-pores that overcome limitations of the skin barrier, thereby significantly enhancing drug penetration. When utilized in combination with gas therapy, MNs exhibit a pronounced enhancement in drug delivery efficiency, surpassing the capabilities of non-gas-producing MNs. The utilization of gas to promote base separation and facilitate drug diffusion is a strategy that has been employed to optimize the safety and efficacy of transdermal delivery. In summary, the present study adopts MNs as its central research focus, leveraging their unique advantages over traditional TDDS while addressing their current shortcomings through combination with gas delivery. Firstly, this review methodically appraises the merits and limitations of MN delivery, then explores the MN-mediated gas delivery for rapid separation and enhanced drug penetration. Subsequently, we investigated the application potential of combined therapy assisted by MNs-mediated gas delivery for various diseases, along with an analysis of their potential developments and future challenges (Figure 1).

## 2. MNs-Mediated Gas Delivery Facilitate Rapid Separation

The existing polymeric MN systems are characterized by inherent limitations. The process of wearing off the MN patches to ensure the release of the therapeutic agent can be quite protracted, often requiring several minutes in the case of water-soluble materials or even several days in the case of biodegradable polymers. This prolonged duration can lead to diminished patient compliance and the occurrence of undesirable cutaneous effects, such as erythema and edema. This phenomenon is of particular concern in the case of pediatric patients who are naturally active and thus less likely to remain still [25,26]. To address these challenges, researchers have enhanced patient comfort by incorporating gas-producing substances within MNs. The gas bubbles that form upon skin penetration facilitate the rapid separation of the MN tip from the substrate, thereby allowing the drug-loaded tip to remain embedded in the skin. This approach has been demonstrated to enhance the efficacy of drug delivery while concurrently reducing the duration of MN contact with the skin [27].

Ning et al. [21] developed a Dual-Layer Dressing Microneedle System (DDMNS). The DDMNS comprised two components: the first component was the chitosan hydrogel dressing (CSHD), which served as the upper layer dressing, and the second component was the detachable MN patch with a panax notoginseng saponins (PNS)-loaded chitosan (CS) tip and a polyvinyl pyrrolidone (PVP) backing substrate loaded with magnesium (Mg) microparticles, which served as the bottom layer dressing. The Mg-PVP backing substrate exhibited rapid dissolution in the humid environment facilitated by the upper layer dressing. Furthermore, the moist local environment provided by CSHD enhances the response between Mg particles and hydrogen ions in an inflammatory environment, thereby facilitating the separation of the MN tip from the substrate. In model wound tissues, MNs containing magnesium achieved a separation rate of 87.5% and demonstrated effective drug penetration within 5 min. In contrast, DDMNS without Mg exhibited a reduced degree of penetration.

Yang et al. [26] (Figure 1A) designed an actively separable MN patch for treating growth hormone deficiency. The researchers integrated the effervescent agent sodium bicarbonate (NaHCO_3_) into a separating poly (acrylic acid) (PAA) layer. Upon absorbing interstitial skin fluid, PAA generates protons, which react with NaHCO_3_ to produce CO_2_. This reaction is characterized by its rapidity and efficacy, resulting in the separation of the MN tip from the base within approximately 11.41 ± 0.43 s (Figure 1C). The experimental results demonstrated a separation efficiency that exceeded 95% in isolated pig skin, in comparison to a mere 32% in the absence of NaHCO_3_ (Figure 1B). In a rat model of growth hormone deficiency, the weekly application of this MN patch demonstrated efficacy that was comparable to the daily subcutaneous injections.

In another study, Li et al. [28] (Figure 1D) introduced a MN patch designed for sustained levonorgestrel (LNG) release. The researchers incorporated effervescent substances, namely NaHCO_3_ and citric acid, between the tip and substrate layers. This effervescent MN patches demonstrated the capacity to achieve rapid MN separation within 10.7 s in phosphate-buffer solution, while MNs without effervescent backing exhibited a longer separation time (Figure 1E,F). Ex vivo experiments also validated this rapid separation, with the separated MN tips remaining embedded in the skin to degrade gradually and sustain LNG release for up to one month in rat models.

## 3. MNs-Mediated Gas Delivery for Enhanced Drug Penetration

MN technology is a promising drug delivery system, with polymeric MNs being most prevalent [29]. However, their drug delivery typically relies on passive diffusion [30], and skin elasticity limits MN penetrate depth [31], often insufficient for effectively treating deep tumors or thickened skin conditions like hypertrophic scar and psoriasis [32,33]. Consequently, strategies to enhance drug delivery efficiency using MNs are required. Although external driving forces such as ion electroosmosis [30], ultrasound [34], or magnetic fields [35] can improve drug diffusion, they necessitate complex equipment, making gas-driven MNs a simpler, more practical alternative.

Gas-producing MNs function by embedding substances that generate gas upon contact with interstitial skin fluid, creating bubbles and vortices that significantly enhance drug diffusion and penetration [36,37]. Typically, effervescent components (organic acids combined with basic carbonate salts) can produce CO_2_ upon hydration. For instance, Zhang et al. [22] used PVP K30, polyvinyl alcohol (PVA), potassium carbonate (K_2_CO_3_), and citric acid to develop gas-driven MNs, achieving a cumulative transdermal drug delivery rate 1.19 times greater than passive MNs. In another study, Chen et al. [37] enhanced anti-follicle aging treatment by employing NaHCO_3_ and citric acid to facilitate deeper drug penetration. You et al. [36] (Figure 2A) similarly developed an ultra-rapid-action MN patch (URA-MN) using NaHCO_3_ and citric acid, rapidly producing CO_2_ bubbles and a porous structure to accelerate drug release within minutes. Frozen sections show that the drug loaded in URA-MN could rapidly diffuse within 5 min (Figure 2B). Wen and colleagues [38] (Figure 2C) also introduced NaHCO_3_ and citric acid into MNs as a built-in engine to generate CO_2_ bubbles, thereby enabling enhanced lateral and vertical drug diffusion into dense scar tissue.

Furthermore, gas generation utilizing NaHCO_3_ has been exploited to target acidic microenvironments [39]. Tao et al. [40] developed a bubble pump MN system, integrating NaHCO_3_ within copper sulfide nanoparticles loaded with fucoidan and doxorubicin in hyaluronic acid MNs. The generated CO_2_ notably improved drug penetration in vitro. Shi et al. [41] created cannabidiol loaded MNs employing calcium carbonate and NaHCO_3_ to produce CO_2_, enhancing cannabidiol penetration and altering tumor microenvironment pH to promote immune responses against melanoma. Ke et al. [42] similarly demonstrated sequential drug release via MNs containing Alexa 488 and polylactic acid-polyglycolic acid hollow microspheres loaded with Cy5 and NaHCO_3_, significantly enhancing drug diffusion under acidic conditions. Addressing postoperative analgesia, Zhang et al. [43] developed a pH-responsive core–shell MN containing ropivacaine microcrystals. Under acidic conditions, CO_2_ production ruptured the MN shell, achieving sustained analgesia for up to 72 h. For diabetes management, Ullah et al. [44] developed MNs encapsulating insulin, glucose oxidase (GOx), and NaHCO_3_ within porous polymer layers coated with polylactic acid-polyglycolic acid films. Upon the diffusion of glucose into the porous layer, the ensuing oxidation to gluconic acid is catalyzed by GOx. This reaction initiates a local pH decline, leading to the decomposition of NaHCO_3_ to yield CO_2_. The glucose-induced CO_2_ release ruptured the polymer film, enabling controlled insulin release and preventing hypoglycemia.

Besides CO_2_, other gases such as O_2_ and H_2_ have been explored. Sodium percarbonate functions as an oxygen-releasing material, facilitating the delivery of sustainable levels of oxygen to hypoxic sites and tissues [45]. As posited by Liu and colleagues [46], the utilization of chlorin e6 (Ce6)-loaded MN patches has been demonstrated to facilitate more profound transdermal drug delivery, in conjunction with enhanced photodynamic therapy (PDT). This enhancement is achieved through the incorporation of sodium percarbonate into PVP MNs. In vivo experiments in tumor-bearing mice demonstrated that tumor growth was significantly inhibited, and the survival rate increased by 50%. Lopez-Ramirez et al. [47] developed degradable MNs with Mg particles generating H_2_, significantly improving drug penetration and immune response in melanoma treatment. Psoriasis is a prevalent chronic inflammatory skin disease marked by accelerated keratinocyte proliferation. Consequently, enhancing drug delivery to the epidermis can markedly improve therapeutic efficacy [48]. Zheng et al. [24] created a microbe-driven bilayer MN employing *Enterobacter aerogenes* (*E.A.*) to generate H_2_, effectively enhancing drug penetration depth. This innovative approach offers precise control of drug delivery depth and provides a promising advancement for MN drug delivery systems.

## 4. MNs-Mediated Gas Delivery for Combined Therapy

Beyond its capacity to expedite the segregation of the tip and base and the diffusion of pharmaceutical agents, gas can also function as a therapeutic tool. Gas therapy has emerged as a novel therapeutic modality, garnering heightened interest due to its notable capabilities in the management and treatment of various diseases [49,50]. This prominence can be ascribed to the merits of gas therapy, including its efficiency and biosafety when compared to conventional pharmaceutical interventions [51]. In clinical practice, therapeutic gases are administered by inhalation. However, the therapeutic potential of this mode of administration is limited by several factors. Direct inhalation of gases is difficult to control, and there may be a risk of gas toxicity [52,53]. In recent years, due to the rapid development of nanomaterials, some researchers have enhanced the efficacy of gas therapy by integrating gases or gas precursors with nanocarriers to enable controlled, responsive release of gases in response to endogenous stimuli (e.g., H_2_O_2_, lactic acid, glucose, and enzymes) and exogenous stimuli (e.g., light, X-rays, ultrasound, magnetic fields, and heat) [49,54]. As a multifunctional platform, MNs can be used for gas therapy through transdermal drug delivery, while combining different therapeutic strategies for synergistic treatment [55]. Advances in medical technology have led to the exploration of MN-mediated gas delivery as a potential therapeutic modality for various diseases. In this section, we provide a synopsis of the various types of therapeutic gases (Table 1) and recent advancements in MN-mediated gas delivery for various diseases (Table 2).

### 4.1. Gas Species for Disease Treatment

#### 4.1.1. Hydrogen (H_2_)

Recent research has demonstrated that low concentrations of H_2_ can neutralize intracellular reactive species such as hydroxyl radicals and peroxynitrites, thereby reducing oxidative stress, inflammation, apoptosis, and fibrosis without disrupting normal metabolic processes [56]. Conversely, high concentrations of H_2_ can induce apoptosis in cancer cells by inhibiting energy metabolism, suppressing vascular endothelial growth factor (VEGF) expression, modulating the PI3K/Akt pathway, and triggering systemic immune responses [57]. However, traditional H_2_ delivery methods, such as inhalation, oral consumption, and injection of H_2_-rich water, encounter significant limitations. These methods lack targeted delivery, owing to H_2_’s high diffusivity and low solubility, which impede its therapeutic efficacy at lesion sites [58,59].

To address this challenge, Yuan et al. [57] developed metal-semiconductor heterostructured nanocomposites (Pt-Bi_2_S_3_), which, under ultrasound irradiation, generated substantial amounts of H_2_ under normoxic or hypoxic conditions. This strategy caused severe mitochondrial dysfunction in tumor cells, disrupted the antioxidant defense system, and effectively induced tumor cell death. Similarly, Gong et al. [60] engineered calcium hydride (CaH_2_) nanoparticles dispersed in polyethylene glycol, which, upon injection into tumors, reacted with water to produce abundant H_2_, calcium ions (Ca^2+^), and hydroxyl ions (OH^−^), enabling H_2_ therapy, intracellular calcium overload, and tumor microenvironment neutralization, collectively suppressing tumor growth. Xu et al. [61] developed a novel near-infrared (NIR)active photocatalyst through the heating of melamine and 2,4,6-triamino-pyrimidine in a solid potassium chloride template. This system-mediated NIR photocatalysis has been shown to generate H_2_, thereby reducing the energy metabolism and contributing to efficient tumor therapy.

#### 4.1.2. Carbon Monoxide (CO)

CO is a colorless, odorless gas historically known for its high affinity for hemoglobin, resulting in toxicity at elevated concentrations, such as hypoxia, convulsions, or death. Recently, however, CO at low physiological concentrations has gained attention as a signaling molecule, demonstrating beneficial biological effects including anti-inflammatory, anti-apoptotic, antihypertensive, vasodilatory, anti-atherosclerotic, and cytoprotective properties. Despite these promising attributes, clinical application of gaseous CO has been hindered by challenges in safe storage, controlled delivery, and targeted release. CO-releasing molecules have emerged as promising alternatives, safely delivering CO to therapeutic sites for conditions such as inflammation, cardiovascular disease, microbial infections, cancer, and organ transplantation [62].

For example, Ma et al. [63] developed a NIR-II activated aggregation-induced emission (AIE)-based nanocarrier, releasing CO specifically in the tumor microenvironment. This released CO inhibited heat shock protein expression, significantly enhancing the efficacy of low-temperature photothermal therapy (PTT). Wang et al. [64] introduced a gaseous nano-adjuvant containing manganese carbonyl encapsulated with AIE luminogens, which released CO under NIR laser irradiation, triggering mitochondrial damage and activating the cGAS-STING pathway, thus enhancing anti-tumor immunotherapy in poorly immunogenic mammary tumors. Additionally, Wu et al. [65] developed engineered exosomes loaded with 5-hexyl aminolevulinate hydrochloride, enabling biosynthetic CO and bilirubin production upon administration, resulting in potent anti-inflammatory effects and improved treatment of atherosclerosis.

#### 4.1.3. Nitric Oxide (NO)

NO is an endogenous gaseous signaling molecule integral to various physiological processes, including vasodilation, regulation of blood pressure and flow, inhibition of platelet aggregation and leukocyte adhesion, smooth muscle proliferation, and neurotransmission [66]. Due to its short half-life and susceptibility to interactions with glutathione (GSH), hemoglobin, superoxide, and molecular oxygen, direct therapeutic use of free NO gas is challenging. To address these limitations, multifunctional nanocarriers containing NO donors or NO-releasing agents have been extensively investigated [67].

For instance, Wang et al. [68] designed photosensitizer-free polymeric nanocapsules loaded with the NO donor DETA NONOate, achieving controlled NO release within acidic lysosomes of cancer cells. Upon photoacoustic activation, reactive oxygen species (ROS) generated at the site reacted with released NO, forming peroxynitrite (ONOO^−^), inducing mitochondrial dysfunction, DNA damage, and subsequent cancer cell death. Zheng et al. [69] developed a biohybrid assembly of carbon-dots doped nitrogen-rich carbon and Escherichia coli bacteria, generating NO from nitrate under controlled irradiation, achieving effective anti-tumor photodynamic microbial therapy. Similarly, Sun et al. [70] fabricated electrospun nanocomposite membranes, producing sufficient ROS and NO under NIR illumination, enabling highly effective antibacterial PDT. He et al. [71] engineered conductive, self-healing hydrogels capable of controlled and targeted NO release under NIR irradiation, significantly enhancing angiogenesis and diabetic wound healing.

#### 4.1.4. Oxygen (O_2_)

O_2_ therapy, particularly hyperbaric O_2_ therapy, is extensively used clinically to treat decompression sickness, arterial embolism, carbon monoxide poisoning, osteomyelitis, radionecrosis, and chronic wounds [72]. O_2_ availability critically influences PDT, as tumor hypoxia limits PDT efficacy. Intravenous oxygenation faces challenges due to oxygen’s poor solubility in blood, necessitating alternative methods for localized oxygen supply [73].

To address this problem, Zhou et al. [74] reported an activatable singlet O_2_-generating system using linoleic acid hydroperoxide and catalytic Fe^2+^ ions under acidic tumor conditions. Jiang et al. [75] designed conjugated polymer nanoparticles linked to hemoglobin, enhancing chemiluminescence resonance energy transfer-mediated ROS production and O_2_ supply for effective anticancer PDT. Furthermore, Chen et al. [76] developed algal gel patches providing sustained O_2_ delivery, effectively healing chronic hypoxic wounds. Guan et al. [77] proposed hydrogels containing O_2_-releasing microspheres and ROS-scavenging systems to continuously supply O_2_, enhancing cell survival, angiogenesis, and tissue regeneration in diabetic wounds.

#### 4.1.5. Hydrogen Sulfide (H_2_S)

H_2_S is an endogenous gaseous signaling molecule recognized for its antioxidant, anti-inflammatory, and cytoprotective properties [78]. Despite these beneficial effects, direct inhalation of H_2_S poses toxicity risks, prompting exploration of nanocarriers and H_2_S-donor systems for controlled therapeutic release.

Giovinazzo et al. [79] employed slow-release H_2_S donor sodium GYY4137 (NaGYY) in Alzheimer’s disease models, significantly reducing cognitive and motor deficits. Wang et al. [64] developed virus-mimicking hollow mesoporous organosilica structures releasing H_2_S in tumor cells upon GSH-triggered degradation, inducing targeted anticancer effects. Xie et al. [80] synthesized FeS-embedded bovine serum albumin nanoclusters that simultaneously released Fe^2+^ and H_2_S under acidic tumor conditions, inhibiting cancer cell peroxidase activity. Xie et al. [81] demonstrated cardioprotective effects of the H_2_S donor GYY4137 during myocardial ischemia–reperfusion injury by reducing oxidative stress. Similarly, Wang et al. [82] showed that exogenous H_2_S alleviated emphysema and airway inflammation via activation of Nrf2 and PPAR-γ signaling pathways.

#### 4.1.6. Carbon Dioxide (CO_2_)

CO_2_ therapy enhances localized blood supply, improves tissue oxygenation, and exhibits antiseptic and anti-tumor properties [83]. Due to systemic administration challenges, researchers have developed innovative delivery methods [84]. Xie et al. [85,86] developed photothermally activated hydrogels and CaCO_3_-based nanoparticles for targeted CO_2_ release, effectively treating skin wounds and tumors. Jeon et al. [87] created immunostimulatory nanoparticles generating CO_2_ and ROS, inducing immunogenic cell death and enhancing anticancer efficacy.

Overall, emerging strategies leveraging gas therapies offer promising advances in cancer treatment, diabetic wound management, wound infection control, and other pathological conditions, highlighting their potential for improved therapeutic outcomes.

**Table 1 pharmaceutics-17-01576-t001:** Gas species for disease treatment.

Gas Type	Gas Carrier or Precursor	Driving Force or Stimulation	Treatment of Disease	Ref
H_2_	Pt-Bi_2_S_3_	$Pt-{Bi}_{2}S_{3}+H^{+}\overset{US}{\to }H_{2}$	Anti-4T1 tumor therapy	[57]
	Nanoscale CaH_2_ particles	$CaH_{2}+H_{2}O\to H_{2}+{Ca}^{2+}+{OH}^{-}$	Anti-orthotopic liver tumor therapy	[60]
	Carbon/potassium doped heptazine-based red polymer carbon nitride (RPCN)	$RPCN\overset{NIR}{\to }H_{2}$	Anti-4T1 tumor therapy	[61]
CO	CORM (donor)	CORM→CO	Treatment of inflammation, cardiovascular disease, microbial infections, cancer	[62]
	Chemiexcitation-triggered AIE nanobomb (PBPTV@mPEG(CO))	$PBPTV@mPEG\overset{Excess\;H_{2}O_{2}}{\to }CO$	Anti-4T1 tumor therapy	[63]
	Gas nano-adjuvant (MTHMS)	$MTHMS\overset{AIEgen}{\to }{Mn}^{2+}+CO$	Anti-4T1 tumor therapy	[64]
	Hydrochloride-containing M_2_ exosomes (HAL@M_2_ Exo)	HAL release→Heme biosynthesis metabolism→CO+bilirubin	Anti-Atherosclerosis	[65]
NO	NO-loaded nanocapsules (NO-NCPs)	$NO-NCPs\overset{H^{+}}{\to }DETA+NO$	Anti-EMT6 tumor therapy	[68]
	Carbon-dot doped carbon nitrid and *E*. *coli MG1655* (CCN@ *E*. *coli*)	$NO_{3}^{-}\overset{-e^{-}}{\to }NO$	Anti-4T1 tumor therapy	[69]
	Electrostatically spun nanocomposite membrane (UCNP@PCN@LA-PVDF)	$LA\overset{ROS\;induction}{\to }NO$	Antimicrobial therapy	[70]
	Carboxymethyl chitosan/2,3,4-trihydroxybenzaldehyde/copper chloride/graphene oxide-N, N′-di-sec-butyl-N, N′-dinitroso-1,4-phenylenediamine hydrogel(CMCS/THB/Cu/GB)	$CMCS/THB/Cu/GB\overset{NIR}{\to }NO$	Diabetic wound healing treatment	[71]
O_2_	Iron oxide-linoleic acid hydroperoxide nanoparticles (IO-LAHP NPs)	$IO-LAHP\overset{H^{+}}{\to }{1O}_{2}$	Anti-U87MG tumor therapy	[74]
	Hemoglobin-nanoparticles @liposomes (Hb-NPs@liposome)	$Hb-NPs\overset{CRET}{\to }1O_{2}$	Anti-tumor therapy	[75]
	Gel beads of *S. elongatus PCC7942*	$CO_{2},CO^{3-},HCO^{3-}\overset{Photosynthesis}{\to }O_{2}$	Diabetic wound healing therapy	[76]
	Oxygen releasing microspheres (ORM)	$ORMs\to O_{2}$	Diabetic wound healing therapy	[77]
H_2_S	NAGYY (donor)	$NAGYY\to H_{2}S$	Alzheimer’s disease treatment	[79]
	MTHMS	$MTHMS\overset{GSH}{\to }GSSG+H_{2}S$	Anti-4T1 tumor therapy	[64]
	Ferrous sulfide-embedded bovine serum albumin nanoclusters (FeS@BSA nanoclusters)	$FeS\overset{H^{+}}{\to }{Fe}^{2+}+H_{2}S$	Anti-Huh7 tumor therapy	[80]
	GYY4137 (donor)	$GYY4137\to H_{2}S$	Anti-atherosclerosis treatment	[81]
	Sodium hydrogen sulfide (NaHS) (exogenous donor)	$NaHS\to H_{2}S$	Relief of emphysema and airway inflammation	[82]
CO_2_	Bicarbonate	$Carbon\;NPs\begin{array}{c} \overset{Sunlight}{\to }\mathrm{Thermal}\;\mathrm{Bicarbonatate}\\ \to {\mathrm{CO}}_{2} \end{array}$	Accelerated wound healing	[85]
	Core–shell hybrid nanoparticles consisting of CaCO_3_ and MnSiO*_x_* (CaCO_3_@MS nanoparticles)	$CaCO_{3}\overset{H^{+}}{\to }CO_{2}$	Anti-4T1 tumor therapy	[86]
	Immunostimulatory CRET nanoparticles (iCRET NPs)	$iCRET\;NPs\overset{H_{2}O_{2}}{\to }CO_{2}$	Anti-CT26 tumor therapy	[87]

### 4.2. Application of MNs-Mediated Gas Therapy

#### 4.2.1. Cancer

Cancer remains a leading cause of premature mortality worldwide [88]. Traditional treatments, including surgery [89], radiotherapy [90], and chemotherapy [91], frequently result in suboptimal outcomes due to drug resistance, systemic toxicity [92], and radiation-induced damage to healthy tissues [93]. Emerging therapeutic modalities, such as immunotherapy [94], targeted therapy [95], and phototherapy [96], have broadened cancer treatment options, yet these approaches also have inherent limitations. Recently, gas therapy has emerged as a promising adjuvant due to its high selectivity, minimal side effects, and distinct therapeutic mechanisms [97,98].

Yang et al. [99] developed a wearable smart MN device composed of a dual-layer MN patch, a heating membrane, a Bluetooth-enabled flexible printed circuit board, a smartphone application, and a wristband. The inner microneedle layer made of silk fibroin (SF) is loaded with anti-PD-1 antibody (aPD-1), while the outer layer is polycaprolactone (PCL) encapsulating ammonia borane-loaded mesoporous silica nanoparticles (AB-MSN). Upon insertion into melanoma lesions and heating to 50 °C via the integrated heating membrane, the outer polycaprolactone coating melts, releasing AB-MSN, which generates H_2_ in the tumor’s acidic environment. H_2_ inhibits tumor cell proliferation and migration and effectively targets and eliminates cancer stem cells. Subsequently, the dissolution of the inner fibroin layer leads to controlled release of aPD-1 antibodies, blocking immune evasion and promoting further tumor destruction by activated T cells. The results of anti-melanoma study demonstrated that the proportion of CD8^+^ T cells in the aPD-1+ AB-MSN group was 38.7% ± 3.1%, which was twice higher than that of the AB-MSN group.

Shi et al. [100] (Figure 3A) fabricated gold-engineered cryomicroneedle (CryoMN) patches containing Rhodospirillum rubrum bacteria (R.r-Au). These bacteria metabolize lactic acid, an immunosuppressive byproduct abundant in tumor microenvironments, to produce H_2_ under laser irradiation. This system effectively transformed tumor-associated macrophages from an M2-type (immunosuppressive) to an M1-type (immunostimulatory) phenotype, restored cytotoxic CD8^+^ T cell activity, and suppressed regulatory T cells. The antitumor effect was verified by measuring the tumor volume and weight of harvested tumors, and the CryoMNs-R.r-Au (+) group showed higher antitumor activity (Figure 3B,C).

Yu and colleagues [101] (Figure 3D) tackled the inefficiencies of chemodynamic therapy (CDT), specifically insufficient tumor H_2_O_2_ and elevated GSH, using MNs loaded with poly(amidoamine) dendrimers functionalized with hydrazine groups, caffeic acid (CA, an H_2_O_2_ enhancer), ferric ions (Fe^3+^, a Fenton reaction initiator), and arginine (an NO precursor). Arginine is converted to NO by tumor-derived H_2_O_2_, simultaneously depleting GSH, thereby markedly enhancing antitumor efficacy in melanoma mouse models. The mice treated with this system exhibited significantly enhanced tumor inhibition compared with other groups (Figure 3E). The tumor growth inhibition rate of against A375 melanoma tumor was calculated to be 94.5% according to the weight of the excised tumors (Figure 3F).

Dong et al. [102] integrated PTT with NO therapy by developing multifunctional MNs containing sodium nitroprusside (SNP) and ferrous lactate (SNP-Fe). Upon activation by ultraviolet (UV) or natural light, SNP released NO, inducing effective tumor ablation at high concentrations while promoting tissue regeneration at lower concentrations. This dual-functionality strategy addressed common limitations of traditional PTT, including inadequate thermal penetration and poor tissue regeneration post-ablation. Mild PTT is a novel therapeutic modality that utilizes temperatures marginally above body temperature (approximately 45 °C) to heat tumor tissues for therapeutic effect, with minimal damage to normal tissues [103,104]. However, mild PTT can result in the excessive expression of heat shock proteins, leading to tumor heat resistance and diminished therapeutic efficacy [105]. To address this issue, Dong and colleagues [106] designed partitioned MNs incorporating copper sulfide nanoparticles and S-nitroso-N-acetylpenicillamine (SNAP). Under irradiation, SNAP produced NO and ROS, suppressing heat shock protein synthesis that typically leads to tumor thermoresistance, thus significantly improving mild PTT therapeutic outcomes.

CO, recognized for its potential as an adjunctive cancer treatment due to its minimal adverse effects, was employed by Jin et al. [107] through hollow mesoporous silica nanoparticles loaded with manganese-based CO-releasing molecules (MnCO@hMSN). These nanoparticles responded to tumor H_2_O_2_ by releasing CO, leading to substantial inhibition of tumor growth with negligible toxicity. Fu et al. [108] further developed core–shell MNs containing NaHCO_3_, tartaric acid, photocatalysts, and doxorubicin hydrochloride. Upon insertion, CO_2_ rapidly generated from the vesicants enhanced the diffusion of doxorubicin hydrochloride and provided substrate for CO release via photocatalysis, significantly enhancing the combined anticancer effects.

In another study, Hu et al. [23] presented smart MNs integrating gas therapy, starvation therapy, and imaging. Optimized Bi/BiVO_4_ Schottky heterojunctions enabled excellent photothermal conversion and computed tomography imaging capabilities, while loaded GOx and diallyl trisulfide (DATS) allowed ultrasound- or NIR-triggered H_2_S gas release, synergistically enhancing antitumor efficacy through combined gas and starvation therapy. In vivo antitumor experiments reH2Svealed that the MN + US + NIR group exhibited the most significant reduction in tumor volume.

#### 4.2.2. Diabetic Wound

Diabetic wound remain clinically challenging, characterized by prolonged hyperglycemia, elevated pH [109], oxidative stress, chronic inflammation [110], impaired angiogenesis, and increased susceptibility to infections [111,112]. Current standard treatments such as surgical debridement and infection control often yield limited success [113,114]. Consequently, gas-based therapies utilizing endogenous signaling molecules, including NO, H_2_S, H_2_, and O_2_, have gained prominence due to their potent roles in modulating inflammation, bacterial infections, and promoting healing [115,116,117,118,119].

Yao et al. [120] (Figure 4A) designed a MN patch composed of copper-based metal–organic frameworks combined with graphene oxide, enabling controlled release of NO under NIR irradiation. This approach significantly accelerated diabetic wound healing by promoting M_2_ macrophage polarization and enhancing angiogenesis (Figure 4B,C). In another study, Cai et al. [121] synthesized a biomineralized nanoenzyme functionalized with Polymyxin B, delivering continuous and controlled release of H_2_S. The nanoenzyme was further loaded into MNs prepared with polyvinyl alcohol and hydroxyethyl methacrylate. These MNs exhibited potent antioxidant, antimicrobial, and anti-inflammatory effects.

Wang and colleagues [122] developed MNs loaded with magnesium hydride (MgH_2_) facilitating sustained H_2_ and magnesium ions (Mg^2+^) release. The ex vivo experiments demonstrated that H_2_ significantly enhanced diabetic wound healing by reducing ROS production as well as Mg^2+^-promoted M_2_ macrophage polarization. In a diabetic mouse model, the wounds in the MN-MgH_2_ group exhibited signs of healing, while the wounds in the other groups remained unresolved. In a similar study, Tao et al. [123] designed a MN patch loaded with graphene oxide (GO)/Co^2+^ and ammonia borane@mesoporous silica nanoparticles (AB@MSN). The H_2_ generated by AB@MSN has been shown to remove excess ROS, and subsequent H_2_ reduction of oxygen-containing functional groups in GO has been demonstrated to result in the release of Co^2+^. This, in turn, has been observed to promote vascular regeneration. In vivo experiments in diabetic rats demonstrated the efficacy of this system in promoting wound healing.

Chronic diabetic wounds are characterized by impaired oxygen delivery to the wound site due to compromised blood vessel integrity. This results in the formation of a hypoxic environment surrounding the wound, which, in turn, contributes to a protracted healing process [124]. To address this issue, Sun et al. [125] (Figure 4D) used silk fibroin methacryloyl (SiLMA) as a MN material, with the tip of the MN encapsulated with calcium peroxide and catalase, and the bottom coated with antimicrobial silver nanoparticles. Calcium peroxide, when exposed to water, undergoes a reaction that results in the formation of H_2_O_2_. In the presence of catalase, the decomposition of H_2_O_2_ occurs, leading to the sustained production of O_2_ for up to seven days. In a murine model of diabetes, the system demonstrated a substantial propensity for enhancing angiogenesis and M_2_ macrophage polarization (Figure 4E,F). As indicated by the works of Ran et al. [126], Zhang et al. [127], Liu et al. [128], Zhao et al. [129], and Gao et al. [130], the focus also has been placed on the alleviation of hypoxia in diabetic wounds through the utilization of MNs for the delivery or generation of oxygen. These innovative strategies incorporated calcium peroxide, catalase, responsive hydrogels, and photosynthetic microalgae (*Chlorella vulgaris*), improving oxygenation, enhancing angiogenesis, and facilitating faster wound healing.

**Figure 4 pharmaceutics-17-01576-f004:**
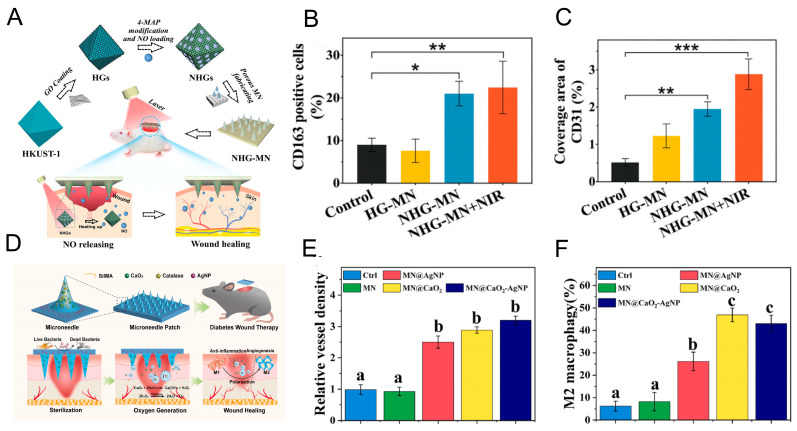
(**A**) Schematic diagram of the preparation and application of the MN array for NO release to promote diabetic wound healing. (**B**) Quantification analysis of CD163 positive cells. (**C**) Quantitative evaluation of the coverage area of CD31. * *p* < 0.05, ** *p* < 0.01, *** *p* < 0.001. Reproduced with permission from ref. [120]. under the terms of the Creative Commons CC BY license. Copyright 2021, with permission from Wiley. (**D**) Schematic illustration of the MN patch for O_2_ generation to promote wound healing. (**E**) Quantitative analysis of vascular density. (**F**) Analysis of the proportion of M_2_ macrophages. Different letters denote statistically significant differences (*p* < 0.05). Reproduced with permission from ref. [125]. under the terms of the Creative Commons CC BY 4.0 license. Copyright 2024, with permission from Elsevier.

#### 4.2.3. Wound Infection

Gas therapy mediated by MNs is also effective against infected wounds. In a recent study, Shi et al. [131] designed a multifunctional heterostructure consisting of ultrasmall platinum–ruthenium nanoalloys and porous graphitic carbon nitride C3N5 nanosheets (PtRu/C3N5), which were embedded within hyaluronic acid MNs. This approach resulted in the generation of H_2_ under light irradiation, accompanied by noteworthy antimicrobial and anti-inflammatory activities. The authors demonstrated that the H_2_ production increased with irradiation time and that there was no significant decrease in H_2_ production efficiency after four consecutive cycles of irradiation. The in vivo experiments demonstrated that this system exhibited a substantial enhancement in the healing process of skin wounds in bacterially infected mice.

Jin et al. [132] (Figure 5A) proposed a novel MN patch loaded with NO-encapsulated nanocarriers for the synergistic treatment of chronic skin wounds. The nanocarriers were composed of sodium alginate (SA) encapsulating reuterin (RE@SA) and immobilized concanavalin A (Con A) and NO molecules (RE@SA-ConA/SNO). Con A can target bacteria through specific recognition of polysaccharides on bacterial surfaces, which is followed by the release of the antimicrobial agent to achieve an effective antimicrobial effect. With the assistance of MNs, the nanoparticles could achieve effective antibacterial effect (Figure 5B). Moreover, NO was released in a sustained manner accompanied by the dissociation of nanoparticles in the wound tissue, which exerted potent anti-inflammatory action and benefited vascular regeneration (Figure 5C), further promoting chronic wounds closure.

In another study, Xu and colleagues [133] (Figure 5D) constructed a multifunctional MN platform encapsulating a ROS-responsive CO donor (CORM-401) and a diagnostic fluorescent probe (TICO). Upon encountering acid-producing bacteria, the probe undergoes molecular structural change and exhibits significant near-infrared fluorescence output for detecting infections and assessing their severity. Concurrently, the nanoprobe is capable of generating ROS, which not only directly eliminates the bacteria through oxidative damage, but also triggers the release of CO for adjuvant therapy. In a mouse subcutaneous infection model, infected wounds treated with TICO@MN plus light exhibited nearly complete healing, with a wound healing rate exceeding 87%. Compared with other experimental groups, the wounds treated with TICO@MN + light groups showed a significant decrease in IL-6 and TNF-a levels (Figure 5E), thereby inhibiting the spread of wound inflammation and promoting tissue regeneration.

#### 4.2.4. Other Diseases

Beyond cancer and wounds, MN-based gas therapy shows promise in treating conditions such as Achilles tendinopathy, traumatic brain injury, acute myocardial infarction, postoperative pain, and skin photoaging.

Achilles tendinopathy (AT) is a chronic degenerative disease in which persistent oxidative stress and inflammation lead to tendon degeneration [134]. Recent studies have demonstrated the efficacy of tendon stem cell-derived exosomes in promoting the healing of injured tendons [135]. However, satisfactory therapeutic effects cannot be obtained by injecting exosomes. To address this issue, Liu et al. [136] (Figure 6A) designed a detachable MN array loaded with tendon-derived exosomes and NO-producing nanomotors for the purpose of facilitating AT healing. L-arginine functions as an NO donor, leading to the production of anti-inflammatory gaseous NO in the presence of ROS. Additionally, NO functions as a driving force, thereby enhancing the depth of penetration of exosomes. The in vivo experiments demonstrated that the MN patch induced a more efficient healing process in AT rats than a single exosome injection.

Traumatic brain injury (TBI) is a series of injuries caused by external head trauma, such as intracranial hematomas, which can lead to disability and even death [137]. Chiang et al. [138] (Figure 6B) proposed the development of high-frequency magnetic field (HFMF)-responsive NO-release gold yarn-dynamos (mNOGO). These dynamos utilize HFMF to trigger NO release and electrical stimulation, with the objective of restoring brain function after TBI. The system consists of an NO donor (poly(S-nitrosoglutathione)) and a gold yarn-dynamo (GY), which is loaded into a silk MN. In vivo experiments demonstrated that the GY and NO donors-loaded MNs lowered astrocytes and immune responses, leading to improved nerve regeneration. Moreover, the MNs promoted angiogenesis with the assistance of HFMF, thereby facilitating the regeneration of neurons in vivo.

**Figure 6 pharmaceutics-17-01576-f006:**
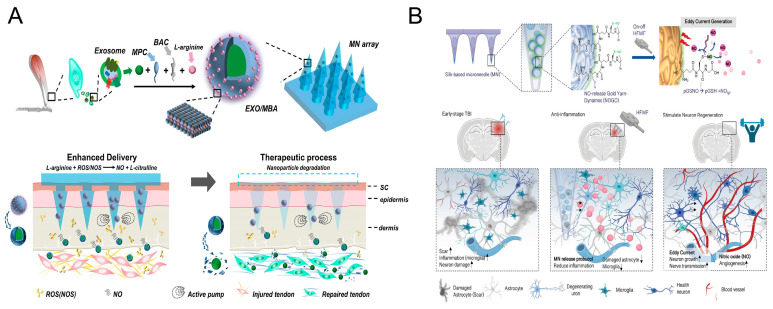
(**A**) Schematic illustration of MN array loaded with tendon-derived exosomes and NO-producing nanomotors for AT healing. (**B**) Schematic illustration of MNs-mediated NO release to improve TBI therapeutic efficacy. Reproduced with permission from ref. [138]. under the terms of the Creative Commons CC BY license. Copyright 2023, with permission from Wiley.

Acute myocardial infarction (AMI) is a serious cardiovascular emergency that arises primarily from coronary artery occlusion. The condition is characterized by an overproduction of ROS, in addition to persistent inflammation and oxidative stress. Wang et al. [139] developed a ROS- and US-responsive bilayer MNs. In the acute phase of AMI, the upper MNs could rapidly release cerium oxide nanoparticles in the presence of ROS. In contrast, during the chronic repair phase, micro-nano reactors (mesoporous silica nanoparticles loaded with curcumin and L-Arg (CLMS)) located in the lower MNs produced NO under ultrasound irradiation. The experimental results indicated that the NO generation was activated by US irradiation. This design has been demonstrated to resist oxidative stress and reduce inflammation in the acute phase, while promoting myocardial repair in the chronic phase. In evaluations of in vivo therapeutic efficacy, the US + MN group demonstrated improvements in cardiac function and exhibited the most potent immunomodulatory effects.

Postoperative pain management is a critical issue that needs to be addressed in modern anesthesiology and perioperative medicine. Inadequate pain management has been demonstrated to prolong hospital stays, impede wound healing, and diminish quality of life. Previous research indicates that H_2_ has the capacity to not only mitigate pain but also to facilitate wound healing by decreasing inflammation and oxidative stress. Zhang et al. [140] employed dissolving MNs to deliver polydopamine-modified ZIF-8@ammonia borane nanoparticles (PDA@ZIF-8@AB) and temperature-responsive QX-314-loaded PCL microspheres. The experimental results demonstrated that the nanoparticles could sustainably release H_2_ in the postoperative acidic environment, alleviating pain and promoting wound healing. Meanwhile, PCL@QX-314 microspheres, activated by NIR light, could regulate the release of the local anesthetic drug QX-314, thereby achieving effective analgesic effects.

Excessive exposure to UV radiation is a major factor in the development of skin photoaging wrinkles. Although there are currently various methods for combating photoaging, including the topical application of antioxidants, Botox injections, and platelet-rich plasma therapy [141,142], the therapeutic effects are often limited to slowing the progression of the disease, making it difficult to achieve true “reversal”. Lin et al. [143] introduced galvanic cell MN patches with magnesium-containing bipolar electrodes. These patches operate through a galvanic cell mechanism, generating microcurrents and releasing H_2_ and magnesium ions via a redox reaction. The combination of antioxidant and anti-inflammatory properties of H_2_, microcurrent-induced stimulation of cell migration, and promotion of angiogenesis and macrophage M_2_ polarization of magnesium synergistically works to reverse photoaging wrinkles and rejuvenate the skin. This approach demonstrates potential for promoting research and development in the domain of medical cosmetology.

**Table 2 pharmaceutics-17-01576-t002:** Applications of MNs-mediated gas delivery.

Application	Gas	MN Types	Gas-Producing Components	Condition	Ref
Facilitate rapid separation	CO_2_	Dissolving MNs	NaHCO_3_	Growth hormone deficiency	[26]
	CO_2_	Dissolving MNs	NaHCO_3_, citric acid	Long-acting contraception	[28]
	H_2_	Dissolving MNs	Mg	Chronic wounds	[21]
Enhanced drug penetration	CO_2_	Dissolving MNs	K_2_CO_3_, citric acid	/	[22]
	CO_2_	Dissolving MNs	NaHCO_3_, citric acid	Androgenetic alopecia (AGA)	[37]
	CO_2_	Dissolving MNs	NaHCO_3_, citric acid	Diabetic	[36]
	CO_2_	Dissolving MNs	NaHCO_3_, citric acid	Hypertrophic scar	[38]
	CO_2_	Dissolving MNs	NaHCO_3_	Melanoma	[40]
	CO_2_	Dissolving MNs	NaHCO_3_	Melanoma	[41]
	CO_2_	Dissolving MNs	NaHCO_3_	/	[42]
	CO_2_	Dissolving MNs	NaHCO_3_	Postoperative pain	[43]
	CO_2_	Coated MNs	NaHCO_3_	Diabetic	[44]
	H_2_	Dissolving MNs	Mg	Melanoma	[47]
Combined therapy	H_2_	Dissolving MNs	AB-MSN	Melanoma	[99]
	H_2_	Dissolving MNs	AB-MSN	Diabetic wounds	[123]
	H_2_	Dissolving MNs	MgH_2_	Diabetic wounds	[122]
	H_2_	Dissolving MNs	E.A.	Psoriasis	[24]
	H_2_	Dissolving MNs	PDA@ZIF-8@AB	Postoperative pain	[140]
	H_2_	Dissolving MNs	PtRu/C3N5	Wound infection	[131]
	H_2_	Galvanic cell MNs	Mg	Skin photoaging	[143]
	H_2_	Cryomicroneedles	R.r-Au	Melanoma	[100]
	NO	Dissolving MNs	P-NO-CA@Fe	Melanoma	[101]
	NO	Dissolving MNs	SNAP	Melanoma	[106]
	NO	Dissolving MNs	SNP-Fe	Maxillofacial malignant skin tumors	[102]
	NO	Dissolving MNs	RE@SA-ConA/SNO	Wound infection	[132]
	NO	Hydrogel MNs	mNOGO	TBI	[138]
	NO	Hydrogel MNs	CLMS	AMI	[139]
	O_2_	Hydrogel MNs	Oxygen-carrying protein hemoglobin	Diabetic wounds	[127]
	O_2_	Hydrogel MNs	Dopamine-functionalized sericin protein (SDA)	Diabetic wounds	[128]
	O_2_	Hydrogel MNs	*Chlorella vulgaris*	Diabetic wounds	[129]
	O_2_	Hydrogel MNs	*Chlorella*	Diabetic wounds	[130]
	O_2_	Hydrogel MNs	Calcium peroxide	Diabetic wounds	[125]
	O_2_	Dissolving MNs	Manganese/Dopamine-enhanced calcium peroxide	Diabetic wounds	[126]
	H_2_S	Dissolving MNs	DATS	Breast cancer	[23]
	H_2_S	Dissolving MNs	Biomineralized nanoenzyme functionalized with Polymyxin B	Diabetic wounds	[121]
	CO	Dissolving MNs	TICO	Wound infection	[133]

## 5. Conclusions

In summary, gas-mediated delivery via MNs demonstrates significant advantages in both drug transport and disease therapy. Medical gases can accelerate MN tip separation to reduce application time and improve patient compliance, while simultaneously serving as driving forces to enhance drug diffusion and penetration through the skin, thereby improving therapeutic efficacy. More importantly, gases such as H_2_, CO, NO, O_2_, H_2_S, and CO_2_ possess intrinsic biological activities that can be effectively delivered through MNs and act synergistically with other therapeutic modalities, including PTT, PDT, CDT, and chemotherapy. Current studies have shown that MN-based gas delivery can administer diverse gas-loaded nanomaterials for the treatment of skin diseases (e.g., melanoma, diabetic wounds, psoriasis) as well as other diseases (e.g., AT, TBI, AMI, postoperative pain, skin photoaging). Furthermore, the integration of MNs with electronic devices is driving the intelligent evolution of transdermal diagnostics and therapeutics, enabling real-time physiological signal monitoring while releasing drugs precisely on demand.

Despite these promising advances, several challenges remain. First, existing research is primarily focused on melanoma, highlighting the need to broaden applications to other skin disorders such as atopic dermatitis and hypertrophic scars. Second, the bidirectional effects of gases at different concentrations underscore the importance of precise dosage control. Most current studies employ in vitro experiments to measure gas production. These experiments use tools like gas chromatography-mass spectrometry, electrochemical sensors, kits, and fluorescence probes. However, it is still hard to monitor how gas is released from MNs in the body in real time. Consequently, the development of a simple and real-time in vivo gas release monitoring system is imperative to ensure its safety and therapeutic efficacy. Third, to ensure stability and prevent premature gas leakage during transportation or storage, it is essential to select appropriate MN materials. Ideally, the MN material should remain completely inert and stable during dry storage and only undergo dissolution or degradation upon contact with target tissue fluid (or under specific triggering conditions), thereby releasing gas as needed. In addition, the industrialization of gas-producing MNs still faces bottlenecks in manufacturing feasibility and scalability. There is an urgent need to address challenges in precision, consistency, and cost during large-scale production through systematic process optimization and automation upgrades. While the majority of MNs are still in the preclinical phase of gas delivery research, a number of MN technologies have advanced to the clinical trial stage, including MN-assisted vaccines and insulin delivery [144]. This finding indicates the potential for clinical translation of the system, contingent upon the resolution of the aforementioned challenges.

Overall, MN-mediated gas delivery not only enhances transdermal efficiency but also expands therapeutic strategies and clinical applications. This innovative approach holds substantial potential for future clinical translation as a next-generation treatment modality.

## Data Availability

No new data were created or analyzed in this study.

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
