# Peer review of "Microneedle Mediated Gas Delivery for Rapid Separation, Enhanced Drug Penetration, and Combined Therapy"

_pharmaceutics, 2025, doi:10.3390/pharmaceutics17121576_

Round 1

Reviewer 1 Report

Comments and Suggestions for Authors

The manuscript is well-written and presents a comprehensive review that is highly relevant to the broader interests and scope of Pharmaceutics. The topic of microneedle-mediated gas delivery is timely and innovative.

The article is publishable after addressing a few minor revisions aimed at maximizing its scholarly impact and clarity.

Specifically

  • Clarity: In the Abstract and major headings, ensure the wording clearly highlights the three core functions of gas delivery (rapid separation, enhanced penetration, and combined therapy).

  • Table Accuracy: In Table 1, verify that the "Driving force" column lists the stimulus and not the product. 

  • Conclusion Nuance: Slightly adjust the conclusion to acknowledge the research covers a wide range of diseases, even if cancer/melanoma remains the most extensively studied area.

Reviewer 2 Report

Comments and Suggestions for Authors

The paper by Quan et al. reviews the recent advances in microneedle (MN) mediated gas delivery systems for transdermal therapeutic applications, with a focus on rapid separation, enhanced drug penetration, and combined therapy. The review outlines the fundamental challenges of conventional MN systems, particularly the slow detachment of MN tips and limited transdermal efficiency due to skin barriers. It highlights recent strategies using gas-assisted MNs, which either generate or release gases such as hydrogen, oxygen, nitric oxide, hydrogen sulfide, carbon monoxide, and carbon dioxide at the site of administration. These gas generating approaches not only expedite MN tip separation, thereby improving patient compliance, but also significantly enhance drug diffusion and penetration. The article extensively discusses therapeutic applications, including cancer, diabetic wounds, and infections, and describes innovative multi-functional MN patches that integrate gas-generating components for synergistic effects. The review also outlines the main limitations of current research. It suggests future directions, emphasizing the need for precise dose control, expansion to broader disease areas, and improved fabrication techniques for next-generation intelligent MN systems.

While the paper is innovative and addresses an important topic, several areas require further attention to strengthen its impact and clarity:

  1. The abstract and introduction would benefit from a more thorough and up to date literature review. Incorporating recent, relevant studies would help position your work within the current field and better identify advances and remaining gaps. For example, consider citing recent publications such as:

https://advanced.onlinelibrary.wiley.com/doi/full/10.1002/adhm.202400881

  1. Provide additional critical assessment of methodological weaknesses in the cited studies, especially regarding in vivo to clinical translation and experimental controls.​
  2. Highly recommend discussing intricacies in controlling and monitoring gas dosage and release kinetics in MN patches, indicating available quantitative methods or highlighting needs for real-time monitoring.
  3. Add a comparative section evaluating gas-mediated MN systems versus other advanced transdermal technologies or non-gas enhanced MNs.
  4. Discuss the stability and shelf life of gas generating MN patches, considering material degradation and gas leakage during transportation/storage
  5. Suggest more potential applications or disease models beyond those already mentioned (e.g., atopic dermatitis, vaccines, chronic pain, or cosmetic uses
  6. Review advances in integrating MNs with electronics (e.g., monitoring, feedback, or controlled release devices) for greater clinical applicability.
  7. Provide a more granular breakdown of future barriers, such as cost, manufacturing upscaling, patient acceptability, and real-world performance.
  8. Incorporate summary tables or schematic diagrams to visually integrate key data, mechanisms, and comparative outcomes of different gas types and MN designs.
